

# Multi-gene phylogeny and divergence estimations for Evaniidae (Hymenoptera)

Barbara J. Sharanowski[1], Leanne Peixoto[2], Anamaria Dal Molin[3] and Andrew R. Deans[4]

[1] Department of Biology, University of Central Florida, Orlando, FL, United States of America
[2] Department of Agroecology, Aarhus University, Aarhus, Denmark
[3] Departamento de Ciências Biológicas, Universidade Federal do Espírito Santo, Vitória, ES, Brazil
[4] Frost Entomological Museum, Department of Entomology, Pennsylvania State University, University Park, PA, United States of America

## ABSTRACT

Ensign wasps (Hymenoptera: Evaniidae) develop as predators of cockroach eggs (Blattodea), have a wide distribution and exhibit numerous interesting biological phenomena. The taxonomy of this lineage has been the subject of several recent, intensive efforts, but the lineage lacked a robust phylogeny. In this paper we present a new phylogeny, based on increased taxonomic sampling and data from six molecular markers (mitochondrial *16S* and *COI*, and nuclear markers *28S*, *RPS23*, *CAD*, and *AM2*), the latter used for the first time in phylogenetic reconstruction. Our intent is to provide a robust phylogeny that will stabilize and facilitate revision of the higher-level classification. We also show the continued utility of molecular motifs, especially the presence of an intron in the *RPS23* fragments of certain taxa, to diagnose evaniid clades and assist with taxonomic classification. Furthermore, we estimate divergence times among evaniid lineages for the first time, using multiple fossil calibrations. Evaniidae radiated primarily in the Early Cretaceous (134.1–141.1 Mya), with and most extant genera diverging near the K-T boundary. The estimated phylogeny reveals a more robust topology than previous efforts, with the recovery of more monophyletic taxa and better higher-level resolution. The results facilitate a change in ensign wasp taxonomy, with *Parevania*, and *Papatuka*, **syn. nov.** becoming junior synonyms of *Zeuxevania*, and *Acanthinevania*, **syn. nov.** being designated as junior synonym of *Szepligetella*. We transfer 30 species to *Zeuxevania*, either reestablishing past combinations or as new combinations. We also transfer 20 species from *Acanthinevania* to *Szepligetella* as new combinations.

# INTRODUCTION

Ensign wasps (Hymenoptera: Evaniidae) are common, nearly cosmopolitan, and include approximately 500 extant species in 21 genera, although many species remain to be described (*Deans, 2005*). Their biology lies at the precipice between wasps that provision their young with prey and parasitic wasps that deposit their offspring to feed on one host. A female evaniid wasp lays a single egg within a cockroach egg case and their offspring feeds on the unhatched cockroach eggs. Because their larvae feed on multiple individuals

Corresponding author
Andrew R. Deans, adeans@psu.edu

ensign wasps are regarded as predators as opposed to parasitoids (*Huben, 1995*). However, the intimate association that larval evaniids have with their prey is much more reminiscent of parasitoid behavior. Despite these interesting biological features, there is scant research aimed at understanding their evolution and natural history. This predicament remains, in part, due to ongoing instability in their classification and the lack of robust diagnostic tools and inadequate taxon descriptions. Taxonomic work over the last 20 years, however, including a key to genera (*Deans & Huben, 2003*), a comprehensive species catalog (*Deans, 2005*, treating all ca. 500 species), descriptions of fossils (*Deans et al., 2004*; *Jennings, Krogmann & Priya, 2013*; *Jennings , Krogmann & Mew, 2012*; *Jennings, Austin & Stevens, 2004*), and updated (*Deans & Kawada, 2008*) and semantically-enhanced species-level revisions (*Balhoff et al., 2013*; *Mikó et al., 2014*) have substantially increased the potential for research on these insects.

*Deans, Gillespie & Yoder (2006)* also published the first phylogeny of the family, which was an attempt to test the historic generic and tribal classifications. Of the 17 included genera, four were represented by single specimens: *Papatuka* Deans, *Rothevania* Huben (monotypic), *Thaumatevania* Ceballos (monotypic), and *Trissevania* Kieffer. Six genera were found to be monophyletic in both a parsimony and Bayesian analysis, including: *Acanthinevania* Bradley, *Decevania* Huben, *Evania* Fabricius, *Evaniscus* Szépligeti, *Micrevania* Benoit, and *Semaeomyia* Bradley. Although *Prosevania* Kieffer was always recovered, with one possibly misplaced specimen of *Szepligetella* Bradley, it is likely that *Prosevania* may also be monophyletic. Several other genera were consistently recovered as paraphyletic or in unresolved polytomies, including *Brachygaster* Leach, *Evaniella* Bradley, *Hyptia* Illiger, *Szepligetella* Bradley, *Parevania* Kieffer, and *Zeuxevania* Kieffer. The latter two genera were consistently recovered in a clade with *Papatuka* Deans, and *Deans, Gillespie & Yoder (2006)* suggested that these taxa may be congeneric based on the molecular results and inconsistencies in the morphological character that separates these two genera (presence of fore wing 1RS in *Parevania*). They also suggested *Evaniella* may be monophyletic as it was consistently recovered with the exception of one aberrant taxon, since described as its own genus (*Alobevania Deans & Kawada, 2008*).

The only tribal classification put forth for Evaniidae was by *Bradley (1908)*, who suggested two tribes for the ten genera described at the time: Hyptiini (including *Evaniella*, *Evaniscus*, *Hyptia*, *Parevania*, *Semaeomyia*, and Zeuxevania) and Evaniini (including Acanthinevania, *Evania*, *Prosevania*, and *Szepligetella*). This tribal classification was not supported by *Deans, Gillespie & Yoder (2006)*. There was not enough resolution to confidently resolve relationships among evaniid genera to develop a better tribal classification. *Deans, Gillespie & Yoder (2006)* did suggest that the New World taxa with reduced wing venation (including *Evaniscus*, *Decevania*, *Hyptia*, *Rothevania*, and *Semaeomyia*) were monophyletic and could represent a tribe.

The poorly resolved phylogenies published by *Deans, Gillespie & Yoder (2006)* may be attributed to low taxonomic sampling, as only 54 ingroup taxa were included, or, more likely, a lack of informative sites in the sequence data. The resulting "backbone polytomy", where higher-level classifications remain elusive, is common in other phylogenies of Hymenoptera that use the same or similar sets of genes (*Dowton & Austin, 2001*; *Mardulyn*

& Whitfield, 1999; Pitz et al., 2007). Divergence times for members of Evaniidae have not been estimated before. Several recent studies on Hymenoptera have estimated stem-age divergences for Evanioidea ranging from 175 Ma to 221 Ma (Ronquist et al., 2012a; Zhang et al., 2015; Peters et al., 2017; Branstetter et al., 2017). Unfortunately, the small sample size for Evanioidea in all of these studies (1–3 exemplars) and uncertainty in phylogenetic relationships of Evanioidea within Hymenoptera resulted in wide confidence intervals around the estimates. Based on all fossils placed within Evanioidea, it is likely that the superfamily diversified in the Middle Jurassic but may have originated as early as the late Triassic (Li et al., 2018).

Here we attempt to gain a better understanding of higher-level relationships among genera and better test the monophyly of genera, using an increased taxonomic and genetic sampling dataset, including a handful of new protein-coding genes. Our intent is to provide a robust phylogeny that will stabilize and facilitate revision of the higher-level classification. We also show the continued utility of molecular motifs, first described for Evaniidae by Deans, Gillespie & Yoder (2006), to diagnose clades and assist with taxonomic classification. Furthermore, we estimate divergence times among evaniid lineages for the first time, using multiple fossil calibrations to understand of the timing of diversification in Evaniidae.

## MATERIALS AND METHODS

### Taxon sampling

A list of taxa and sequences utilized in this study is presented in Table 1 (more details in Table S1). Exemplars were obtained for 89 evaniid specimens, across 17 genera, and five outgroup taxa, including two species of *Gasteruption* (Gasteruptiidae) and three species of *Pristaulacus* (Aulacidae), for a total of 94 taxa. All evaniid genera were represented except four rare genera: *Afrevania*, *Brachevania*, *Thaumatevania*, and *Vernevania*. We were only able to include one representative of *Alobevania* and *Rothevania* (monotypic), and *Papatuka*. Where possible, sampling was increased for genera that were previously recovered as paraphyletic by Deans, Gillespie & Yoder (2006).

Each exemplar not identified to species represents a putative morphospecies, as many species remain undescribed. Several DNA extracts and some sequences were used from Deans, Gillespie & Yoder (2006), as indicated in Table 1. Vouchers were deposited at the Frost Entomological Museum, at The Pennsylvania State University, or in repositories stipulated by collecting permits and/or loan agreements (Supplemental Information 1).

### Gene selection

We utilized DNA from six different genes, including two mitochondrial (mt) genes (16S ribosomal DNA (*16S*) and cytochrome c oxidase I (*COI*) and four nuclear genes (28S ribosomal DNA (*28S*), ribosomal protein S23 (*RPS23*), carbamoyl-phosphate synthetase-aspartate transcarbamoylase-dihydroorotase (*CAD*) and alpha-mannosidase II (*AM2*). Diagrams of the gene structures of *CAD*, *RPS23*, and *AM2* are presented in Fig. 1. The diagrams were produced based on annotations of the genomic reference sequences from *Apis mellifera* Linnaeus, 1758 (NCBI RefSeq ID: GCF_000002195.4) and *Nasonia vitripennis* (Ashmead, 1904) (NCBI RefSeq ID: GCF_000002325.3), visualized

**Table 1  Taxonomic and genetic sampling.** Exemplars used by *Deans, Gillespie & Yoder (2006)* are listed with the reference from that paper (DV#) beside the internal voucher number (Ext.). Genes for each taxon are marked with an X if amplified in this study and D if amplified by *Deans, Gillespie & Yoder (2006)*. Gene codes: 28S, 28S rDNA; AM2, alpha-mannosidase II; CAD1 and CAD2, carbamoyl-phosphate sythetase-asparate transcarbamoylase-dihydroorotase (CAD) (for amplicon regions for each segment, see Fig. 1); RPS23, Ribosomal Protein S23; COI, cytochrome oxidase I; and 16S, 16S rDNA.

| Taxon | Ext. | DV# | 28S | AM2 | CAD1 | CAD2 | RPS23 | COI | 16S |
|---|---|---|---|---|---|---|---|---|---|
| *Gasteruption* 300 | 300 | | X | | X | X | X | X | D |
| *Gasteruption* 244 | 244 | | X | | X | | X | X | |
| *Pristaulacus strangaliae* | 176 | | | X | | | X | X | |
| *Pristaulacus fasciatus* | 299 | | | | | | | X | |
| *Pristaulacus* 21 | 306 | 21 | D | | | | X | D | D |
| *Acanthinevania* 240 | 240 | | X | X | X | X | X | X | |
| *Acanthinevania* 242 | 242 | | X | X | X | X | X | | |
| *Acanthinevania princeps* | 246 | | X | | X | X | X | X | |
| *Acanthinevania* 001 | 271 | 001 | D | X | X | X | X | D | D |
| *Acanthinevania* 033 | 289 | 033 | D | X | X | X | X | D | D |
| *Acanthinevania* 049 | 292 | 049 | D | X | X | X | X | D | D |
| *Alobevania gattiae* | 200 | 039 | D | X | X | X | | X | D |
| *Brachygaster minutus* | 273 | 030 | X | | X | X | X | D | D |
| *Brachygaster minutus* | 512 | | | | X | X | | X | |
| *Brachygaster* 037 | 286 | | D | | X | | X | | D |
| *Brachygaster* 050 | 290 | | D | | | | X | | D |
| *Decevania* 502 | 502 | | | | X | X | | | |
| *Decevania* 513 | 513 | | | X | | | | X | |
| *Decevania* 004 | 274 | 004 | D | X | | X | X | D | D |
| *Decevania* 005 | 301 | 005 | D | | | | X | | D |
| *Decevania* 063 | 296 | 063 | D | X | X | X | X | D | D |
| *Evania* 175 | 175 | | X | | | | X | X | |
| *Evania albofacialis* | 275 | 020 | D | | X | X | X | D | D |
| *Evania appendigaster* | 207 | 046 | D | | X | X | X | D | D |
| *Evania* 496 | 496 | | X | | | X | X | X | |
| *Evania* 002 | 189 | 002 | D | | X | X | | D | D |
| *Evaniella* 230 | 230 | | X | X | X | X | X | X | |
| *Evaniella* 234 | 234 | | X | X | | X | X | X | |
| *Evaniella* 237 | 237 | | | X | X | | | X | |
| *Evaniella* 485 | 485 | | | X | X | X | X | | |
| *Evaniella* 486 | 486 | | | X | X | | X | | |
| *Evaniella* 493 | 493 | | | | X | | X | X | |
| *Evaniella semaeoda* | 220 | 058 | D | | | | X | D | D |
| *Evaniella* 019 | 192 | 019 | D | X | X | | X | D | |
| *Evaniella* 025 | 307 | 025 | D | | X | | X | D | D |
| *Evaniella* 045 | 206 | 045 | D | X | X | | X | | D |
| *Evaniscus marginatus* | 213 | 052 | D | | | | X | | D |

*(continued on next page)*

Table 1 (*continued*)

| Taxon | Ext. | DV# | 28S | AM2 | CAD1 | CAD2 | RPS23 | COI | 16S |
|---|---|---|---|---|---|---|---|---|---|
| *Evaniscus rufithorax* | 206 | | D | | | X | X | X | D |
| *Hyptia* 232 | 232 | | | X | X | X | X | X | |
| *Hyptia* 487 | 487 | | | X | | | | X | |
| *Hyptia* 501 | 501 | | | X | X | | | X | |
| *Hyptia* 511 | 511 | | | X | X | | | X | |
| *Hyptia amazonica* | 235 | | | | X | X | | X | |
| *Hyptia floridana* | 291 | 009 | D | X | X | X | | D | D |
| *Hyptia* 007 | 302 | 007 | D | | X | | X | D | D |
| *Hyptia* 008 | 303 | 008 | D | | | | X | D | D |
| *Micrevania difficilis* | 283 | 006 | D | | X | X | | X | D |
| *Micrevania* 061 | 288 | 061 | D | X | | X | | D | D |
| *Micrevania* 066 | 298 | 066 | D | | | X | | D | D |
| *Micrevania* 026 | 308 | 026 | D | | | D | | D | D |
| *Papatuka capensis* | 227 | 065 | D | | X | X | X | X | D |
| *Parevania* 172 | 172 | | X | X | X | | X | | |
| *Parevania* 174 | 174 | | X | X | | | X | X | |
| *Parevania* 041 | 295 | 041 | D | X | X | X | X | D | D |
| *Parevania* 057 | 219 | 057 | D | X | X | X | X | | D |
| *Parevania* 064 | 276 | 064 | D | | X | X | X | D | D |
| *Prosevania fuscipes* | 224 | 062 | D | | | X | | X | D |
| *Prosevania* 497 | 497 | | X | X | X | X | | X | |
| *Prosevania* 498 | 498 | | | | X | | | X | |
| *Prosevania* 508 | 508 | | | | | | | X | |
| *Prosevania* 027 | 309 | 027 | D | X | | X | | X | D |
| *Prosevania* 034 | 277 | 034 | D | | | | | D | D |
| *Prosevania* 036 | 284 | 036 | D | | X | X | | X | D |
| *Prosevania* 044 | 205 | 044 | D | X | X | X | X | D | D |
| *Rothevania valdivianus* | 239 | 048 | D | X | X | X | | D | D |
| *Semaeomyia* 489 | 489 | | | | X | X | X | X | |
| *Semaeomyia* 509 | 509 | | X | | | X | X | X | |
| *Semaeomyia* 510 | 510 | | X | | | X | X | X | |
| *Semaeomyia leucomelas* | 305 | 016 | D | | | X | X | D | D |
| *Semaeomyia* 012 | 197 | 012 | D | | X | X | | D | D |
| *Semaeomyia* 051 | 279 | 051 | D | | | X | X | D | D |
| *Semaeomyia* 059 | 293 | 059 | D | X | | X | X | D | D |
| *Szepligetella* 170 | 170 | | | | X | | X | X | |
| *Szepligetella* 231 | 231 | | X | | X | X | X | X | |
| *Szepligetella* 233 | 233 | | X | X | X | X | X | X | |
| *Szepligetella* 236 | 236 | | X | X | X | X | | X | |
| *Szepligetella* 238 | 238 | | X | X | X | X | X | X | |
| *Szepligetella* 241 | 241 | | | | X | X | X | X | |
| *Szepligetella* 243 | 243 | | X | X | X | | X | X | |
| *Szepligetella* 247 | 247 | | | | X | | X | X | |

**Table 1** (*continued*)

| Taxon | Ext. | DV# | 28S | AM2 | CAD1 | CAD2 | RPS23 | COI | 16S |
|---|---|---|---|---|---|---|---|---|---|
| *Szepligetella 248* | 248 | | X | | X | | X | X | |
| *Szepligetella sericea* | 297 | | | | | X | X | X | |
| *Szepligetella 047* | 208 | 047 | D | X | X | | X | D | D |
| *Szepligetella 055* | 280 | 055 | | X | X | | X | D | D |
| *Szepligetella 056* | 294 | 056 | D | X | X | X | X | X | D |
| *Szepligetella 285* | 285 | | | X | X | X | X | X | |
| *Trissevania anemotis* | 282 | 038 | D | X | | X | X | D | D |
| *Trissevania 507* | 507 | | | | | X | | | |
| *Zeuxevania 499* | 499 | | | | X | X | | | |
| *Zeuxevania 500* | 500 | | | | X | X | | X | |
| *Zeuxevania 503* | 503 | | | | | X | | | |
| *Zeuxevania 505* | 505 | | X | | X | X | X | X | |
| *Zeuxevania 015* | 191 | 015 | D | | X | | | D | D |
| *Zeuxevania splendidula* | 312 | 031 | D | | X | X | X | X | D |
| % amplified | | | 71 | 44 | 66 | 64 | 67 | 86 | 50 |
| % parsimony-informative | | | 40 | 44 | 49 | 53 | 35 | 60 | 40 |

in NCBI's Sequence Viewer (http://www.ncbi.nlm.nih.gov/tools/sviewer) and Geneious v.6.0.6 (Biomatters Ltd.) The annotations include information on the introns, exons, organization of coding regions and protein product features. Conserved domains in the protein products were also identified via a BLASTx search (*Altschul et al., 1990*) against NCBI's Conserved Domains Database (CDD) (*Marchler-Bauer et al., 2015*). The genetic regions corresponding to the identified domains are included for reference in the diagrams as well as the primers used in this study (primer sequences are listed in Table S2). Further background about the three protein coding genes is provided below since the amplified regions or genes utilized are novel for phylogenetic studies. All sequences are available in NCBI's Genbank (https://www.ncbi.nlm.nih.gov/genbank/) under accession numbers KY082187–KY082565.

### CAD

*CAD* is a long and complex gene which codes a "fusion" protein, that is, a protein with multiple enzymatic activities: glutamine-amidotransferase (GATase), carbamoylphosphate synthetase (CPSase), dihydroorotase (DHOase) and aspartate/ornithine transcarbamoylase (ATCase/OTC). There are 26 exons and 25 introns in both *Apis* and *Nasonia*, although intron loss has been reported in the CPSase small chain region in some Braconidae (*Sharanowski, Dowling & Sharkey, 2011*). CPSase is divided in two domains: one for a short chain, which includes GATase, and one for a long chain. The long chain is also subdivided, consisting of two subunits (N-terminal + ATP-binding region), one oligomerization domain, and one MGS-like (methylglyoxal synthetase-like) domain. These two CPSase chains are coded by 14 exons. Various segments of this gene have been used in other phylogenetic studies of insects, particularly for lineages diversifying within the last 150 million years (*Danforth, Fang & Sipes, 2006*; *Moulton & Wiegmann, 2004*; *Winterton & De Freitas, 2006*).

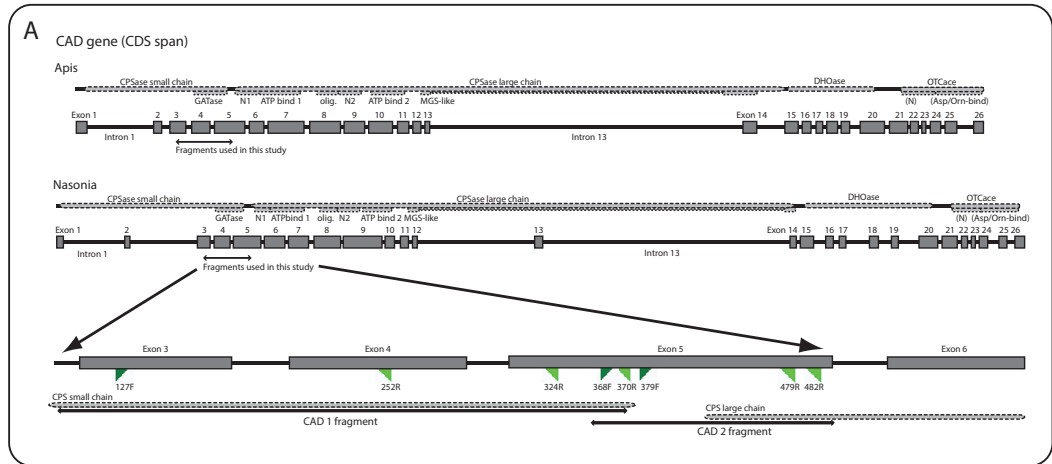

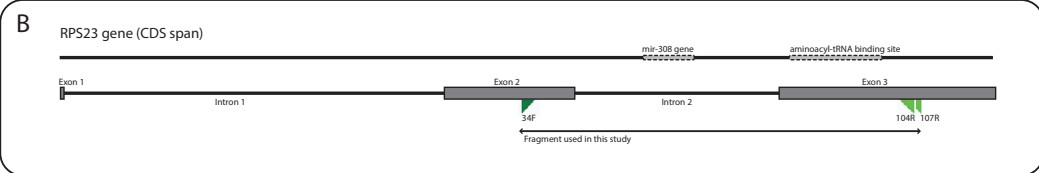

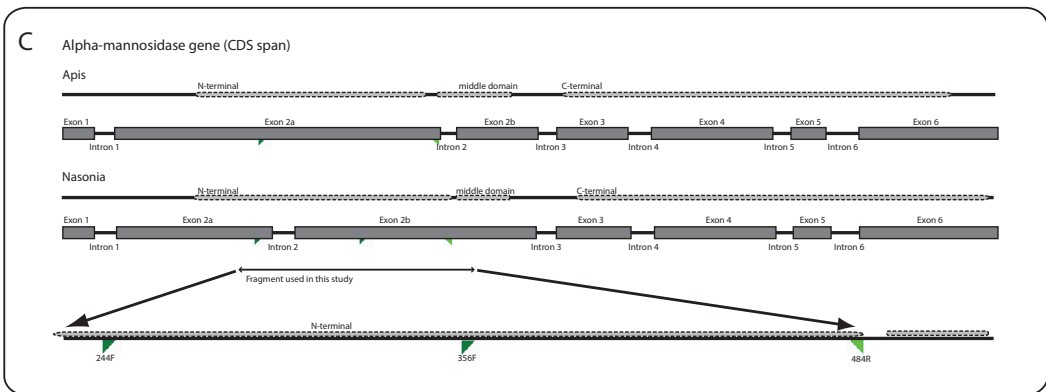

**Figure 1** **Diagrammatic gene maps for: (A) carbamoyl-phosphate sythetase-asparate transcarbamoylase-dihydroorotase (CAD); (B) ribosomal protein S23 (RPS23); and (C) alpha-mannosidase II (AM2). Dotted lines mark protein domains and features. For CAD and AM2, Apis and Nasonia gene diagrams are shown individually as references due to substantial differences in exon locations.** The bottom diagram in each gene map depicts the regions amplified in this study. In *CAD*, intron 13 in *Nasonia* has been scaled down due to an incomplete sequence in the GenBank entry. Primers are named according to the amino acid position in the *Apis mellifera* protein. Forward primers are in dark green and reverse primers in light green. See Table S2 for primer combinations. Abbreviations: *CPS*, carbamoyl-phosphate synthase; *GAT*, glutamine aminotransferase; *DHO*, dihydroorotase; *MGS*, methylglyoxal-like; *OTC*, ornithine carbamoyltransferase; *SN1*, N-terminal of subunit 1 in CPS large chain; *N2*, N-terminal of subunit 2 in CPS large chain; olig., oligomerization domain.

The regions we analyzed are within the CPSase domains, extending between exons 3 to 5 (Fig. 1A).

### RPS23

Ribosomal protein S23 (Fig. 1B) is part of the small ribosomal subunit (*Wool, 1979*). It has a binding site for mRNA and is associated with the eukaryotic initiation factor of the translation process (NCBI-CDD:cd03367). This gene has been previously used in macro-evolutionary phylogenetic studies on Hymenoptera (*Sharanowski et al., 2010*) and Arthropoda (*Aleshin et al., 2009*; *Timmermans et al., 2008*) and as an EPIC (exon-primed, intron-crossing) marker for population-level studies (*Lohse et al., 2011*; *Lohse, Sharanowski & Stone, 2010*). RPS23 is well conserved in sequence and structure across Hymenoptera, with the variation concentrated in the introns. In both *Apis* and *Nasonia*, there are three exons (3 bp, 159 bp, and 270 bp in length) and 2 introns (339 bp and 84 bp in Apis; 353 bp and 79 bp in *Nasonia*). The amplified region covers the downstream region of exon 2, full intron 2, and about half of exon 3, which contains the aminoacyl-tRNA interaction site and therefore is expected to be conserved.

### AM2

We performed sequence similarity searches with tBLASTx (*Altschul et al., 1990*), using Hymenopteran expressed sequence tags (ESTs) from *Sharanowski et al. (2010)* against proteins of *Apis mellifera* and *Nasonia vitripennis*. Our search focused on genes with regions of variability (for putative phylogenetic signal), limited introns and relatively long exons, and regions of sequence conservation (for priming sites). Alpha-mannosidase II is a glycoside hydrolase involved in the catabolism of carbohydrates (*Gonzalez & Jordan, 2000*) and has not been explored for phylogenetic studies. There has been a shift in the placement of the second intron between *Nasonia* and *Apis* (Fig. 1C), and thus we labeled the exons 2a and 2b to demonstrate the homology with labeled exon 3 in both taxa. Three main protein domain regions are identifiable in the reference sequences: (1) an N-terminal catalytic domain of Golgi alpha-mannosidase II, which is entirely in exon 2a in *Apis*, but overlaps the second intron in *Nasonia*, and therefore also lies in exon 2b; (2) a middle domain, which is located in exon 3; and (3) and a C-terminal, which is located in exon 4 (Fig. 1C). The amplified area is contained in the region that corresponds to the N-terminal in *Apis*, ending before the second intron (Fig. 1C). No intron was amplified in the evanioid taxa used in this study, and thus the gene structure is more similar to *Apis* in the amplified region.

## Extraction and sequencing

Extraction of genomic DNA was performed following the manufacturer's protocols using the DNeasy Tissue Kit (Qiagen, USA). Exemplars were either whole body extracted or only the separated thorax and metasoma were used as the use of the head often resulted in low DNA concentrations in Evaniids. *COI* was amplified using the protocols outlined in *Schulmeister, Wheeler & Carpenter (2002)*, with the primers developed for that study or using the universal primers developed by *Folmer et al. (1994)* and following protocols outlined in *Namin, Iranpour & Sharanowski (2014)* (Table S2). Sequences for

16S mitochondrial rDNA were used from *Deans, Gillespie & Yoder (2006)*, which were based on primers and protocols developed in previous studies (*Dowton & Austin, 1994*; *Whitfield, 1997*). Amplification of the D2-D3 region of *28S* was performed using either primers developed by *Dowton & Austin (2001)* or primers newly developed for this study (Table S2), due to difficulty with amplification of some taxa. *CAD* sequences were amplified in two discontinuous fragments using newly developed primers (Fig. 1; Table S2: *CAD1*, *CAD2*). For *CAD1*, three reverse primers were developed to either reduce degeneracy or due to amplification difficulties in some taxa, and a touchdown protocol was also used to increase specificity of the reaction (Table S2). For *CAD2*, two sets of primers were developed, the second set (CAD-Amel379F/CAD-Amel479R) slightly internal to the first (CAD-Amel368F/CAD-Amel482R). If no amplification product was achieved with the first set of primers, the second set was used alone or as a nested re-amplification of the product obtained with the first set. *RPS23* was amplified using primers developed by *Lohse et al. (2011)* and in conjunction with a second newly developed reverse primer and amplified with a touchdown protocol (Fig. 1B; Table S2). Primers were also designed to amplify AM2, with an internal forward primer (AM2-Amel356F) amplifying a much shorter fragment (Fig. 1C, Table S2), which increased the number of taxa for which we achieved amplification success.

All PCR amplifications were carried out using 0.2–1 µg DNA extract, 1 × Standard Taq Buffer (New England Biolabs, USA) (10 mm Tris-HCl, 50 mm KCl, 1.5 mm MgCl2), 200 µm dNTP, 4 mm MgSO4, 400 nm of each primer, 1 unit of Taq DNA polymerase (New England Biolabs, USA) and purified water to a final volume of 25 µL. PCR products were visualized on a 1% agarose gel. Occasionally 5% Dimethyl sulfoxide (DMSO, Sigma-Aldrich, USA) was added as a PCR additive when non-specific bands occurred. This additive has been shown to increase PCR yield with GC-rich templates (*Farell & Alexandre, 2012*). Nested re-amplifications were performed using 0.5 µL of PCR product as DNA template (concentrations varied depending on first PCR reaction success). PCR purification was performed using ExoSAP-IT (Affymetrix, USA) following the manufacturer's instructions, except using 25% of the suggested reagent amount. If double bands were visualized on the gel following PCR, a subsequent 50 µL reaction was run on gel cut bands, the product ran on a 2.5% agarose gel, and purified using the QIAquick Gel Extraction Kit (Qiagen, USA) following the manufacturer's protocols. Sequencing was carried out using the BigDye Terminatorv 3.1 Cycle Sequencing Kit (Applied Biosystems, U.S.A.), with reaction products sequenced on an Applied Biosystems 3 730 ×l DNA Analyzer at the Genomic Sciences Laboratory, North Carolina State University. Contigs were assembled and trimmed for quality using Geneious.

## Sequence alignment

The protein-coding genes were aligned by translating the sequences and setting the correct reading frame in BioEdit (*Hall, 1999*). Sequences were then aligned as proteins using MAFFT (*Katoh & Standley, 2013*) on the EMBL-EBI webserver (*Li et al., 2015*) under default settings and then back translated to nucleotides. Introns present in *CAD1* and *RPS23* were excluded from the dataset prior to multiple sequence alignment. Ribosomal

DNA sequences were aligned following secondary structure models developed by *Gillespie et al. (2005)*; *Gillespie, Yoder & Wharton (2005)* and modified by *Deans, Gillespie & Yoder (2006)* for Evaniidae. Regions of ambiguous alignment (RAA), expansion and contraction (REC), and slipped-strand compensation (RSC) were excluded from the analysis, following *Deans, Gillespie & Yoder (2006)*. For analysis of sequence motifs, introns were aligned using MAFFT with a gap opening penalty of 2 and gap extension penalty of 0.5 to limit excessive gaps in the alignment.

## Phylogenetic analyses

The optimal partitioning scheme and models of evolution for the concatenated analysis were determined using PartitionFinder v.1.1.1 (*Lanfear et al., 2012*). Character sets were predefined by gene, and by codon position for the 5 protein-coding genes for a total of 17 partitions (*CAD1* and *CAD2* were partitioned separately). The Bayesian information criterion was used to select among models implemented in MrBayes version 3.2 *Ronquist et al. (2012b)*, with the greedy search algorithm and branch lengths unlinked. The optimal scheme included two partitions. The first partition included the 3rd codon positions for *CAD1*, *CAD2*, *AM2*, and *RPS23* under the Hasegawa-Kishino-Yano model (*Hasegawa, Kishino & Yano, 1985*). The remaining 13 predefined partitions were included together under the general time reversible model (GTR). Both partitions included a parameter for invariant sites and rate heterogeneity modeled under a gamma distribution. We observed notable differences in nucleotide composition across taxa for some genes (calculated in MEGA v.6 *Tamura et al., 2013*), and thus, tested for base composition homogeneity using chi-square tests in PAUP* (*Swofford, 2002*) (Table S3). For *CAD1* and *RPS23* the intron was removed.

Phylogenies were estimated using MrBayes 3.2, either on the CIPRES Science Gateway (*Miller, Pfeiffer & Schwartz, 2010*) or the ComputeCanada WestGrid computational facility. Parameters were unlinked and site specific rates were allowed to vary across partitions. Analyses were performed with two independent searches and four chains. All concatenated analyses were run for 10 million generations, sampling every 2000th generation. Individual gene trees were analyzed with 5 million generations, sampling every 1000th. Convergence diagnostics, stationarity, and appropriate mixing were assessed with Tracer v1.6 (*Rambaut & Drummond, 2009*), and a suitable burn-in was chosen based on the parameter values. Trees from the posterior distribution were summarized post burn-in with a majority rule consensus and manipulated for better visualization using FigTree v.1.3.1 (*Rambaut, 2012*) and modified for publication using Adobe Illustrator (Adobe Systems, Inc. San Jose, CA). The final nexus file is available through Penn State's ScholarSphere repository (DOI: 10.18113/S1D06H).

## Divergence time estimations

An uncorrelated log-normal relaxed clock as implemented in the program BEAUTi and BEAST v.1.8.2 (*Drummond et al., 2002*; *Drummond et al., 2012*) was used to estimate divergence times. The same partitions and models of molecular evolution were applied to each partition as in the phylogenetic analysis. We utilized the Birth-Death process for

incomplete sampling (*Stadler, 2009*) and started with a random tree. Only the calibration for the entire ingroup (Evaniidae) was constrained to be monophyletic which was well supported from the Bayesian analysis.

We utilized six fossil calibration points with each fossil assigned to the crown group for which they belonged (see Supplemental Information 2) (*Brues, 1933*; *Nel, Martínez-Delclòs & Azar, 2002*; *Pealver et al., 2010*; *Jennings, Austin & Stevens, 2004*; *Jennings, Krogmann & Priya, 2013*; *Jennings & Krogmann, 2009*; *Rasnitsyn, Jarzembowski & Ross, 1998*; *Sawoniewicz & Kupryjanowicz, 2003*). We performed two separate analyses to examine uncertainty with respect to maximum bounds for clade ages. For the first analysis we used log-normal distributions. The age of the fossil determined the hard minimum bound, as the clade to which it belongs must be at least that old. We then chose a mean and standard deviation so that the 95% highest priority density interval (95% HDP) for the divergence estimation of the clade was from 2 to 25 million years prior to the age of the fossil. The 25 million year demarcation is arbitrary, but it seems reasonable and follows *Cardinal & Danforth (2013)*. For the second analysis we chose hard maximum bounds based on previous knowledge of the fossil record and the evolutionary relationships among the included taxa, which are justified (Supplemental Information 2) for each calibration. Generally, we chose the mean as the average between the hard minimum and maximum bounds and then set the standard deviation so that the 95% HDP spanned the range from the minimum to the maximum bound. For both analyses, initial values were set to the mean and the ucld.mean prior was set to exponential with a mean of 0.05. Although these values are somewhat arbitrary, according to the authors of the program, they are unlikely to have an effect on the analysis (*Drummond & Rambaut, 2007*; *Drummond et al., 2012*). All other parameters and the Markov-chain Monte Carlo settings were left at the default settings. The xml input files for both the lognormal and normal distribution analyses are available through Penn State's ScholarSphere repository (DOI: 10.18113/S1D06H).

## RESULTS AND DISCUSSION

### Phylogenetic analyses

The final concatenated data set consisted of 3,097 characters total: *COI* (681 bp), *16S* (371 bp, excluding RAAs), *28S* (428 bp, excluding RAAs), *AM2* (672 bp), *CAD1* (417 bp, excluding the intron), *CAD2* (321 bp), and *RPS23* (207 bp, excluding the intron). Individual gene trees are depicted in Figs. S1–S7. The null hypothesis for base composition homogeneity was rejected for *AM2* ($\chi 2 = 368.819$, *df* $= 120$; $P = 0.000000000$) and *COI* ($\chi 2 = 562.535$, $P = 0.0000000$) (Table S4). Average nucleotide composition across all genes and gene regions analyzed are depicted in Table S4.

The Bayesian analysis of the concatenated dataset recovered a well resolved tree with most clades well supported (pp > 0.95) (Fig. 2). Clades recovered across the individual gene trees and for the concatenated analysis are summarized in Table S4 and gene trees are depicted in Figs. S1–S7. We also performed a Maximum Likelihood analysis with RaxML v8.2.4 (Stamatakis, 2014; *Stamatakis, 2006*) (Fig. S15) under the GTR+CAT model and auto-determination of bootstrap replicates. The phylogenies obtained from BEAST (Fig. 3),

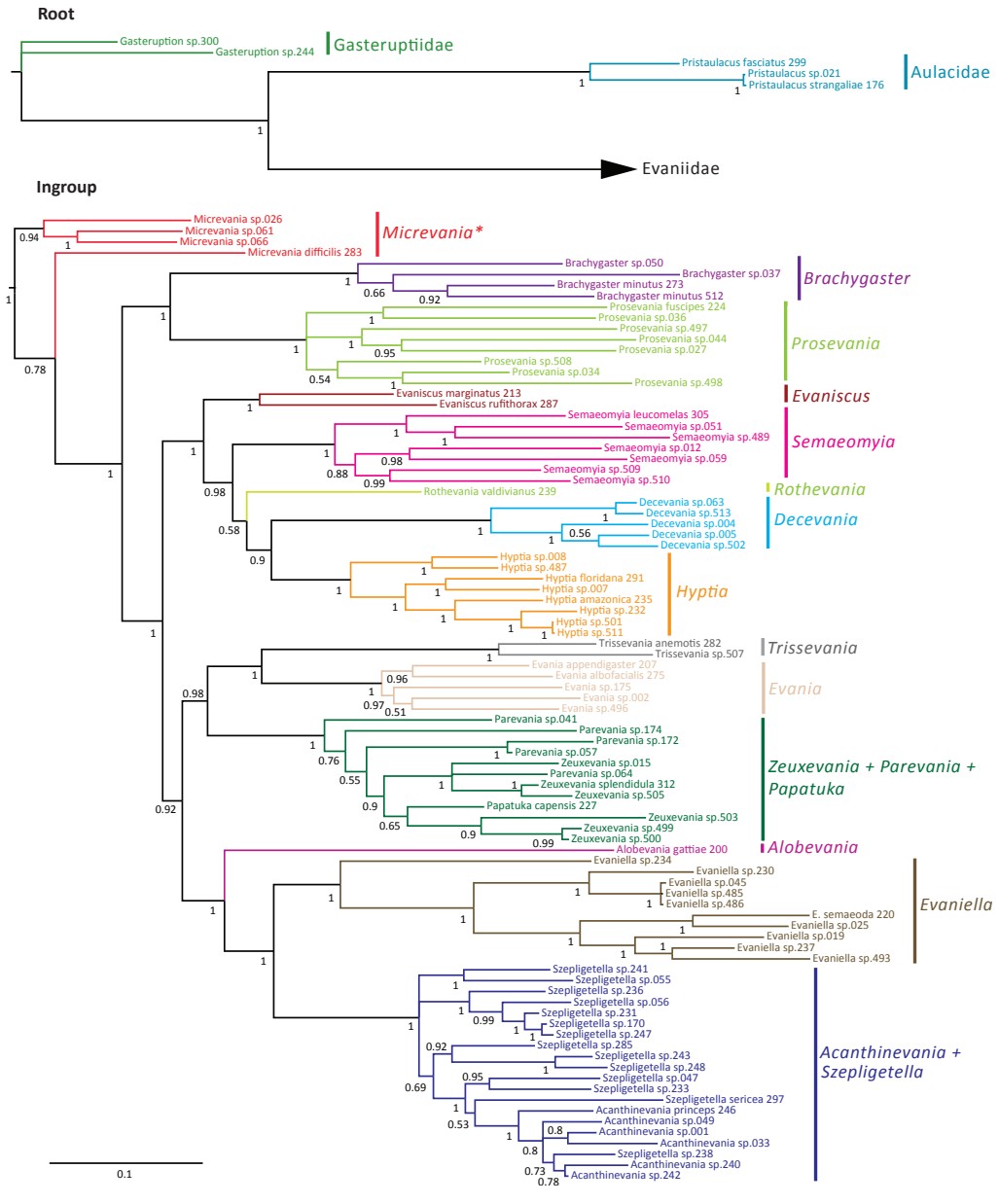

**Figure 2  Bayesian analysis of phylogenetic relationships among Evaniidae.** The outgroups were removed and placed above the ingroup tree for better visualization (the scale has been retained). Posterior probabilities are listed beside each clade.

Mr.Bayes (Fig. 2), and RaxML (Fig. S15) were very similar except relationships among species varied within genera and *Micrevania* was not monophyletic in the Bayesian analysis (Fig. 2). The placement of *Rothevania* also varied across analyses.

In the concatenated analysis (Fig. 2), Evaniidae was recovered as monophyletic with high support (pp = 1.0). Of the 15 genera included in the analysis with more than one representative, nine were recovered as monophyletic, including *Evaniscus*, *Decevania*,

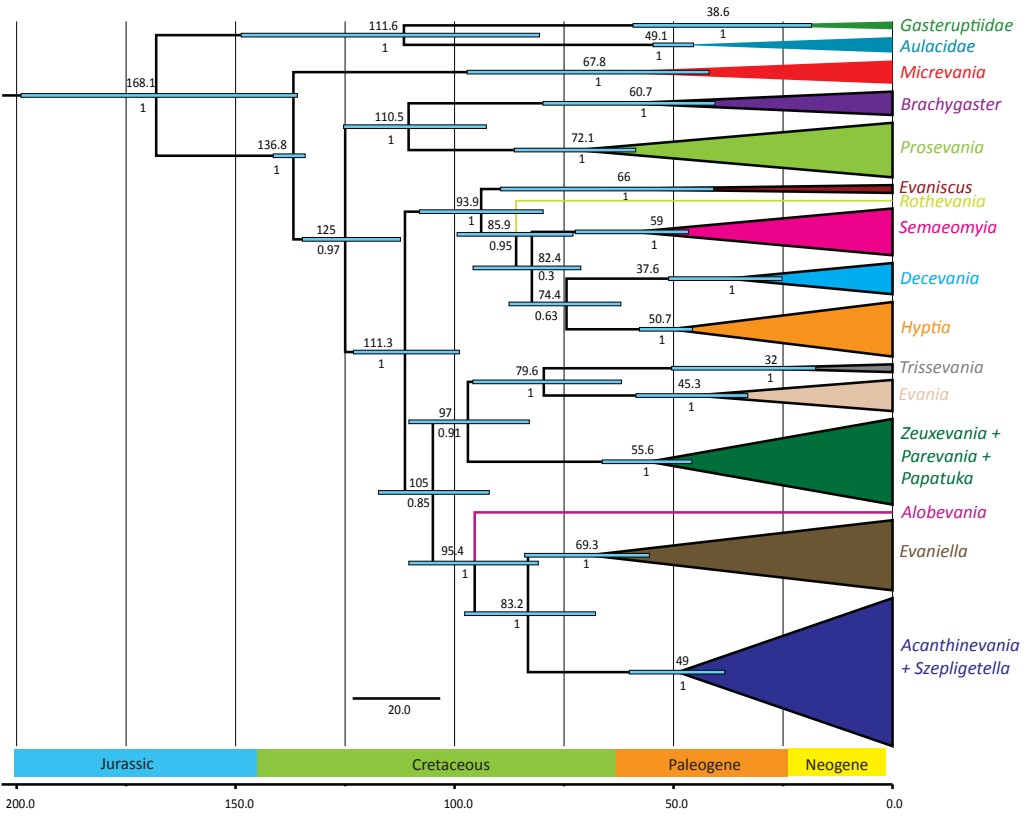

**Figure 3** **Simplified chronogram showing estimated divergence times for Evaniidae with six fossil calibrations and maximum clade ages under a lognormal distribution.** Monophyletic genera have been collapsed for better visualization of the divergence estimations of the major clades. The blue bars indicate the 95% highest posterior density interval (HDP, also listed in Table 1). The scale is in millions of years. Mean age is listed above each clade and posterior probabilities are listed.

*Semaeomyia*, *Evania*, *Hyptia*, *Brachygaster*, *Prosevania*, *Trissevania*, and *Evaniella*. All clades representing monophyletic genera had posterior probabilities of 1.0. Although *Micrevania* was recovered as paraphyletic, it was recovered as monophyletic in other analyses, as mentioned above, the divergence analysis (Fig. 3), ML analysis (Fig. S15) and the *16S* and *COI* individual gene analyses (Table S4) and previously by *Deans, Gillespie & Yoder (2006)*.

Similar to the previous study (*Deans, Gillespie & Yoder, 2006*), *Parevania* and *Zeuxevania* were recovered as paraphyletic with respect to each other, but in a well-supported clade (pp = 1.0) with *Papatuka*, in the concatenated analysis as well as five of the seven gene trees (Table S4). Interestingly, all of these taxa have a distinct sequence motif at the 3′ end of the *RPS23* intron: GTTTGTTTTGYAG (Fig. S9). No other evaniid taxa have a similar motif at the 3′ end (Fig. S8), and thus the motif is diagnostic for this clade. *Trissevania* and *Evania* were recovered as sister taxa with high support (pp = 1.0) in the concatenated analysis and these two taxa were recovered as sister to *Zeuxevania + Parevania + Papatuka* (pp = 0.98) (Fig. 2). But there was little support for these higher level relationships among in the individual gene trees (Table S4). *Brachygaster* was recovered as sister to

*Prosevania* with strong support (pp = 1.0) but was only recovered in the *CAD2* gene tree (Fig. S4). *Micrevania* was also recovered as the sister to all remaining evaniids, followed by *Brachygaster* + *Prosevania* in the concatenated analysis. Yet, the position of these taxa fluctuated widely among the individual gene trees, likely due to inconsistent taxon sampling across the gene trees.

*Acanthinevania* and *Szepligetella* were consistently recovered together (pp = 1.0 in the concatenated analysis (Fig. 2) and all gene trees except *16S* (Table S4), but were paraphyletic with respect to each other. Interestingly, all members of *Acanthinevania* and *Szepligetella* have a GATCTAAC motif (Fig. S10) in the *RPS23* intron that is not shared with any other evaniid taxa (Fig. S8), highlighting their close evolutionary relationship. There are also two diagnostic motifs within regions of ambiguous alignment (RAAs) that were excluded from the phylogenetic analyses. All members of *Acanthinevania* and *Szepligetella* have the motif TAAAAT in RAA8 (Fig. S11) and the motif TGCAYT within RAA12 (Fig. S12). *Evaniella* was recovered as the sister group to *Acanthinevania* + *Szepligetella* in the concatenated analysis and in three genes trees (Table S4). Members of all three genera share a 9 bp diagnostic motif in RAA10 in *28S*: YTCGAWAAA (Fig. S12). Most other evaniid taxa do not have this many base pairs in this position (usually 2–4 bp); the ones that do have longer motifs are radically different in sequence (the full alignment is available in Scholarsphere, DOI: 10.18113/S1D06H). *Alobevania* was recovered as sister to *Evaniella* + (*Acanthinevania* + *Szepligetella*), with strong support in the concatenated analysis, and with moderate support (pp = 0.88) in the *28S* gene tree. This result is unsurprising given that these taxa were once treated as *Evaniella* (*Deans & Huben, 2003*).

New world taxa with reduced wing venation (*Evaniscus*, *Decevania*, *Hyptia*, *Rothevania*, and *Semaeomyia*) are recovered together in a well-supported clade (pp = 1.0), in the concatenated analysis (Fig. 2). This clade is only recovered in the *CAD1* gene tree (Fig. S4), possibly due to lower taxonomic sampling in some individual gene trees due to failed amplification. However, these taxa are present in various combinations throughout the individual gene trees, but the relationships among taxa fluctuate widely, which is also reflected in the lower support values in the concatenated tree for relationships among these genera (Fig. 2).

## Divergence time analyses

The phylogenies obtained from the two Bayesian uncorrelated relaxed clock analyses using BEAST were both identical (Fig. 3 (simplified chronogram from the log-normal distribution) and Fig. S14 (normal distribution)). Other than slight differences among species within genera, and the recovery of *Micrevania* as monophyletic, the trees were very similar to the tree obtained from the analysis with MrBayes (Fig. 2). Estimates of divergence time from both analyses, using either a log-normal and normal distribution are listed in Table 2. The log-normal analysis estimated younger divergence times for all clades (Table 2). This was expected as the calibration bounds were constrained within 25 million years of the fossil's age in the log-normal analysis, but were allowed to vary across a larger span of time in the normal distribution analysis based on interpretations of the fossil record. It is likely that the normal analysis uses too broad a range, with the

**Table 2  Estimates of divergence times for Evaniidae (bolded) and outgroups based on an uncorrelated log-normal relaxed clock analyses.** Six fossil calibrations were used (see Supplemental Information 2) with maximum bounds for clade ages set using a log-normal (Analysis 1) and normal distribution (Analysis 2). For each analysis the mean age in millions of years (My) and the 95% highest posterior density interval (HDP, equivalent to a confidence interval) is provided.

| | Log-normal - Age (My) mean (95% HDP) | Normal - Age (My) mean (95% HDP) |
|---|---|---|
| *Gasteruption* (Gasteruptiidae) | 38.6 (18.5–59.3) | 46.3 (27.2–69.4) |
| *Pristaulacus* (Aulacidae) | 49.1 (45.4–54.7) | 48.9 (23.3–73.6) |
| **Evaniidae** | **136.8 (134.1–141.4)** | **151.5 (135.9–166.7)** |
| *Brachygaster* | 60.7 (40.5–86.4) | 72.1 (49.5–96.3) |
| *Decevania* | 37.6 (25.2–51.1) | 47.8 (31.6–64.0) |
| *Evania* | 45.3 (33.3–58.6) | 55.4 (40.5–70.7) |
| *Evaniella* | 69.3 (55.5–84.0) | 88.6 (73.5–104.3) |
| *Evaniscus* | 66.0 (40.8–89.5) | 80.4 (50.1–110.3) |
| *Hyptia* | 50.7 (45.7–57.8) | 65.4 (50.7–81.2) |
| *Micrevania* | 67.8 (38.4–94.8) | 80.1 (52.5–111.8) |
| *Prosevania* | 72.1 (58.6–86.4) | 85.7 (67.6–103.8) |
| *Semaeomyia* | 59.0 (46.6–72.5) | 76.9 (61.1–94.0) |
| *Szepligetella s.l.* | 49.0 (38.3–60.1) | 60.6 (48.6–72.1) |
| *Trissevania* | 32.0 (17.5–50.4) | 38.5 (20.5–57.3) |
| *Zeuxevania s.l.* | 55.6 (45.8–66.3) | 76.0 (59.6–92.2) |

maximum bound being set too far away from the oldest known fossil for the crown lineage, and thus we depict the log-normal analysis (Fig. 3) and use these dates to draw inferences about evaniid clade divergence. Evaniidae was estimated to diverge around 137 million years ago (Mya) (134.1–141.1). Although the superfamily was not the focus of this study, Evanioidea had an estimated mean age of 168 Mya (135.9–199.0), consistent with other previous estimates suggesting Evanioidea diverged in the mid-late Jurassic (*Peters et al., 2017*; *Branstetter et al., 2017*). Branches leading to *Micrevania*, *Prosevania*, and *Brachygaster* split sometime around the end of the Cretaceous, with means ranging between 60–73 Mya (Table 2). Other extant genera likely diverged sometime in the early Cenozoic and these lineages were likely all present before the start of the Neogene (Fig. 3, Table 2).

## Novel genes and molecular signatures

*Alpha-mannosidase 2* (*AM2*) has never been used before in phylogenetic studies. This gene has a mix of conserved and variable sites (44% parsimony-informative sites), but it failed the test for base composition homogeneity, which can cause systematic bases in phylogenetic analyses (*Phillips, Delsuc & Penny, 2004*; *Rodríguez-Ezpeleta et al., 2007*; *Sharanowski, Dowling & Sharkey, 2011*). RY-coding this gene did not change the results obtained from the concatenated analysis. Unfortunately amplification of *AM2* was difficult, even with the addition of PCR additives such as DMSO, causing a high amount of missing data. Gel cuts were often necessary to achieve clean sequences for several genes, but particularly *AM2*. *RPS23* was highly conserved in the exonic regions, and thus may be better suited for deeper level studies across families. There were distinct molecular

signatures within the intronic region that would be very useful for lower level studies, such as across species, or population-level studies (see *Lohse et al., 2010*). The molecular motifs in the *RPS23* intron were useful for delimiting genera and diagnosing congenerics (see taxonomic implications, below). The individual gene trees for both regions of *CAD* were relatively well resolved (Figs. S4–S5) and similar to other studies (*Desjardins, Regier & Mitter, 2007*; *Sharanowski, Dowling & Sharkey, 2011*), which demonstrates good utility for resolving phylogenetic relationships in Hymenoptera.

Alignments based on secondary structure for rDNA have been very useful for delimiting highly variable regions to exclude from analyses to achieve better phylogenetic results (*Gillespie, Yoder & Wharton, 2005*; *Pitz et al., 2007*). However, variable regions have useful information with phylogenetic and taxonomic utility, as demonstrated by *Sharanowski, Dowling & Sharkey (2011)*, who included variable regions (RECs, RAAs, and RSCs) if the variation in sequence length had a standard deviation less than one. Here we demonstrate the utility of some of these regions for diagnosing genera (Figs. S11–S12) and use these data to improve taxonomic classifications (see Taxonomic implications below).

## Taxonomic implications

Relative to the *Deans, Gillespie & Yoder (2006)* study, the addition of several more genes and taxa clearly led to increased resolution. For example, an additional four genera were recovered as monophyletic, and higher level relationships were more resolved and better supported. Our understanding of evaniid relationships remains incomplete, but, based on mounting evidence here and through our observations of morphology, we feel comfortable proposing the following classificatory changes.

### *New synonyms of Zeuxevania and new combinations*

*Parevania*, **syn. nov.**, and *Papatuka*, **syn. nov.**, are congeneric with and junior synonyms of *Zeuxevania*. *Bradley (1908)* also suspected that these two taxa were congeneric and treated *Parevania* as a subgenus of *Zeuxevania*. These taxa are consistently recovered together in well-supported clades across individual gene trees and within the concatenated analyses, but are polyphyletic with respect to each other (Table S4). Additionally, there are molecular signatures within the *RPS23* intron that support their shared evolutionary history (Fig. S9). ARD has observed thousands of specimens of these taxa and can find no consistency in the presence or absence of the fore wing vein 1RS, which was the only character purported to separate *Parevania* and *Zeuxevania* (*Deans & Huben, 2003*).

Following the taxonomy of *Hedicke (1939)*, we hereby transfer the following species back to *Zeuxevania*: *albitarsus* (Cameron, 1899); *annulicornis* (Turner, 1927); *atra* (Kieffer, 1916); bisulcata (Kieffer, 1911); *curvicarinata* (Cameron, 1899); *kriegeriana* (Enderlein, 1905); *leucostoma* (Kieffer, 1910); *longicalcar* (Kieffer, 1911); *punctatissima* (Kieffer, 1911); *rubra* (Cameron, 1905); *sanguineiceps* (Turner, 1927); schlettereri (Bradley, 1908); *schoenlandi* (Cameron, 1905); *semirufa* (Kieffer, 1907).

We also transfer the following species to *Zeuxevania* for the first time: *aurata* (Benoit, 1950), **comb. nov.**; *brevis* (Brues, 1933), **comb. nov.**; *broomi* (Cameron, 1906), **comb. nov.**; *emarginata* (Kieffer, 1911), **comb. nov.**; *kasauliensis* (Muzaffer, 1943), **comb. nov.**;

*laeviceps* (Enderlein, 1913), **comb. nov.**; *madegassa* (Benoit, 1952), **comb. nov.**; *meridionalis* (Cameron, 1906), **comb. nov.**; *micholitzi* (Enderlein, 1905), **comb. nov.**; *ortegae* (Ceballos, 1966), **comb. nov.**; *plana* (Benoit, 1952), **comb. nov.**; *producta* (Brues, 1933), **comb. nov.**; *remanea* (Brues, 1933), **comb. nov.**.

*Papatuka* was originally described from a single, apterous specimen (*Deans, 2002*) and was since expanded to include other, winged species (*Deans, 2005*). The morphology of these species, which is also reflected in the molecular data, is not substantially different from *Zeuxevania*, and we transfer those species to *Zeuxevania*: *alamunyiga* (*Deans, 2002*), **comb. nov.**; *capensis* (Schletterer, 1886), **comb. nov.**; *longitarsis* (Kieffer, 1904), **comb. nov.**

### New synonym of Szepligetella and new combinations

There is also abundant evidence to support *Acanthinevania* as congeneric with *Szepligetella*. They are consistently recovered together in a clade but neither appears to be monophyletic by itself (Table S4). The primary diagnostic characters that separated these two primarily Australian genera include: *Szepligetella* with the third labial palpomere swollen; *Acanthinevania* with an elongated head relative to *Szepligetella*; and *Acanthinevania* with labium folded strongly anteriorly and thus appearing long and narrow, not broad and flat as in most *Szepligetella* (*Deans & Huben, 2003*). Our observations of more than 1,000 specimens reveal that these character states (e.g., face long vs. face short) fall along phenotypic gradients, with no discrete sets of states. Several molecular characters link (but do not separate) these genera, including motifs present in the *RPS23* intron and at least two regions of 28S Figs. S10–12).

We treat *Acanthinevania*, **syn. nov.**, as *Szepligetella* and transfer the following species to *Szepligetella*: *australis* (Schletterer, 1886), **comb. nov.**; *braunsi* (Kieffer, 1911), **comb. nov.**; *braunsiana* (Kieffer, 1911), **comb. nov.**; *clavaticornis* (Kieffer, 1911), **comb. nov.**; *erythrogaster* (Kieffer, 1904), **comb. nov.**; *eximia* (Schletterer, 1886), **comb. nov.**; *genalis* (Schletterer, 1886), **comb. nov.**; *humerata* (Schletterer, 1889), **comb. nov.**; *leucocras* (Kieffer, 1911), **comb. nov.**; *longigena* (Schletterer, 1889), **comb. nov.**; *lucida* (Schletterer, 1889), **comb. nov.**; *mediana* (Schletterer, 1889), **comb. nov.**; *princeps* (Westwood, 1841), **comb. nov.**; *quinquelineata* (Kieffer, 1904), **comb. nov.**; *rufiventris* (Kieffer, 1911), **comb. nov.**; *scabra* (Schletterer, 1889), **comb. nov.**; *sericans* (Westwood, 1851), **comb. nov.**; *striatifrons* (Kieffer, 1904), **comb. nov.**; *szepligeti* (Bradley, 1908), **comb. nov.**; *versicolor* (Kieffer, 1904), **comb. nov.**; *villosicrus* (Kieffer, 1904), **comb. nov.**

### Emerging tribal classification

A new tribal classification for Evaniidae is warranted, given the lack of support for Bradley's *1908* original (>100 year-old) tribal concepts (*Deans, 2005*; *Deans, Gillespie & Yoder, 2006*; *Deans & Huben, 2003*). *Mikó et al. (2014)* recently described Trissevaniini, to include *Trissevania* and *Afrevania*, and, based on our results here, molecular work by (*Deans, Gillespie & Yoder, 2006*), and prior morphological work by us and our colleagues (*Deans & Huben, 2003*; *Deans & Kawada, 2008*; *Kawada & Azevedo, 2007*; *Kawada, 2011*) we have an opportunity to revise Hyptiini to include those New World genera with reduced wing venation: *Evaniscus*, *Hyptia*, *Rothevania*, *Semaeomyia*, and *Decevania*. We remove

*Brachygaster*, *Evaniella*, and *Zeuxevania* from Hyptiini (see *Bradley, 1908*). This updated concept of Hyptiini can be separated from other evaniid taxa by the absence of at least the fore wing RS+M, and usually many other apical veins (see Figs. 1, 9, 11, 16, 17 in *Deans & Huben, 2003*), and its origin in the New World.

### Evaniid divergence and evolution

Evaniids diverged in the Early Cretaceous (ca. 134.1–141.1 Mya), when numerous modern cockroach fossils have been found (*Grimaldi & Engel, 2005*), although cockroaches with oothecae are thought to have much earlier origins in the Late Carboniferous (*Legendre et al., 2015*). Most of the extant evaniid genera diverged sometime near the K-T boundary, which may indicate that the mass extinction played a role in the divergence of multiple new lineages of ensign wasps. Whether or not there has been co-cladogenesis with modern cockroach lineages remains to be tested but would be hampered by the lack of known host relationships for most evaniids (*Deans, 2005*). For evaniids, as for most Hymenoptera, basic natural history research is needed to understand the trophic relationships among wasps and their hosts.

## CONCLUSION

We provide here a more robust and well-resolved phylogeny for Evaniidae than previous studies, which will facilitate ongoing evolutionary and taxonomic work. Indeed, the new synonyms and combinations proposed above help us progress towards a stable classification that reflects evolutionary relationships. Building on prior results (*Deans, Gillespie & Yoder, 2006*), our data also reveal new, useful markers for Hymenoptera (*AM2* and *RPS23*) and continue to support the utility of shared molecular motifs in defining major clades in Evaniidae. Our results indicate that Evaniidae diverged in the early Cretaceous with most genera diversifying in the late Cretaceous or early Tertiary. The results also highlight important targets for future data collection, especially near the base of the tree (*Micrevania*) and the relationships within each genus. More intensive sampling, especially with the addition of morphological data and fossils (e.g., *Ronquist et al., 2012a*), is the logical next step in providing a tribal classification and more refined estimates for divergence times.

## ACKNOWLEDGEMENTS

This material is based on work supported by the US National Science Foundation (grant numbers DBI-0850223, DEB-0842289, DEB-0956049, and EF0905606). Any opinions, findings, and conclusions or recommendations expressed in this material are those of the authors and do not necessarily reflect the views of the National Science Foundation. We are grateful to Mike Irwin, Martin Hauser, Kevin Holston, Jack Longino, Mike Sharkey, the late Evert Schlinger, the late Don Webb, and countless other collectors who shared material with us. We also thank the numerous loan facilitators, without whom this work would be impossible.

### Funding

This material is based on work supported by the US National Science Foundation (grant numbers DBI-0850223, DEB-0842289, DEB-0956049, and EF0905606). The funders had no role in study design, data collection and analysis, decision to publish, or preparation of the manuscript.

### Grant Disclosures

The following grant information was disclosed by the authors:
US National Science Foundation: DBI-0850223, DEB-0842289, DEB-0956049, EF0905606.

### Competing Interests

The authors declare there are no competing interests.

### Author Contributions

- Barbara J. Sharanowski conceived and designed the experiments, performed the experiments, analyzed the data, prepared figures and/or tables, authored or reviewed drafts of the paper, approved the final draft.
- Leanne Peixoto analyzed the data, prepared figures and/or tables, authored or reviewed drafts of the paper.
- Anamaria Dal Molin analyzed the data, prepared figures and/or tables, authored or reviewed drafts of the paper.
- Andrew R. Deans conceived and designed the experiments, performed the experiments, contributed reagents/materials/analysis tools, prepared figures and/or tables, authored or reviewed drafts of the paper, approved the final draft.

### DNA Deposition

The following information was supplied regarding the deposition of DNA sequences:
The sequences are available at GenBank: accession numbers KY082187 to KY082565.

### Data Availability

Relevant input files for all analyses are available through Penn State's ScholarSphere repository (https://scholarsphere.psu.edu): https://doi.org/10.18113/S1D06H (collection of alignment and other input files).

### Supplemental Information

Supplemental information for this article can be found online at http://dx.doi.org/10.7717/peerj.6689#supplemental-information.

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
