# Peer review of "Multi-gene phylogeny and divergence estimations for Evaniidae (Hymenoptera)"

_PeerJ, doi:10.7717/peerj.6689_

## Round 0.1 · original submission · Major Revisions

As you will see, we were able to count on three experts in the field, who collectively provide a wealth of constructive feedback.

Most points pertain to presentation and missing information, which should be easy to address.

In addition, reviewer 1 expresses pertinent concerns regarding the timing, which may require an additional control to assess the robustness of the estimations.

In any case, please address all the points in earnest.

·

Basic reporting

The language of the article is sufficient, but the writing could be improved to make the article more concise. I am not a native English speaker and thus hesitate to make major corrections, but still suggest the authors check again for typos and aim to improve the writing. There are expressions such as "belongs for a group" which I at least have never heard before.
The taxonomic context is outlined in a very detailed fashion. However, I miss a few more sentences about the diversity and biology of the group, which would make the article more accessible for non-specialists. Context is missing altogether for the dating part, both considering the methodology and previous estimates for Hymenoptera and the evaniid fossil record. The abstract seems to have an overly long introduction and methods part, while the reported results are purely taxonomic, with no mention whatsoever of the dating results and a complete absence of anything resembling a discussion of the implications of this study for the understanding of evaniid evolution.
The authors should also make additional effort to improve figure readability (especially the phylogeny would profit from some colour), and the layout of the supplementary files needs to be improved, given that this will not be done by the journal.
I think you should move a (potentially shortened) list of taxa to the main article, as taxon sampling is so central to taxonomic and phylogenetic studies.
The sequence data has been uploaded on Genbank and input files for the various phylogenetic and dating analyses are available on a repository. However, the file names there are somewhat cryptic. Please add a ReadMe file or similar which explains the contents of the different files and add file extensions where missing.

Experimental design

The Evaniidae certainly could do with a more robust phylogenetic hypothesis than what has been available until now, and the rich fossil record does indeed call for a dating analysis – the work thus would certainly fill a gap. However, I have some major concerns about the execution of this study.
My first major concern is the large number of included specimens that have only been identified to genus. They appear in the tree and tables with numbers that obviously reflect extraction numbers (e.g., "Acanthinevania sp.240"). In fact, only a minority of the terminals have been identified to species, and it is entirely unclear if the remaining ones represent species not yet named, nor if they (within genus) all represent the same (morpho)species or not. Given that the focus of this paper is almost exclusively taxonomic, this is somewhat disconcerting. We know from experience that sequence data (and with it phylogenetic insights) tend to become meaningless if no species name can be attached to it, even if the depository of the voucher is reported in the paper. Is it really not possible to obtain better identifications of these specimens? Are there plans to describe the new species at some point in the near future?
My second issue is with the dating analysis, especially with the choice and interpretation of calibration fossils. You state that you used them "with each fossil assigned to the crown group for which they belonged" (line 240/241). Did you really place these calibrations on the MRCA of the respective group? How can you just assume that they belong to the crown group and are not stem-group representatives? I am not familiar with most of these fossils, but researched a bit on one example: Hyptia deansi (Jennings et al., 2012) has been associated with this genus based mostly on a reduced wing venation. In the original description, the authors state that the distinct notauli distinguished the species from most extant representatives of the genus – could that not be an indication that H. deansi would actually attach to the stem of the genus, instead of one of the crown group branches? Misrepresenting fossils as crown instead of stem groups is known to lead to sometimes massive overestimation of divergence times in molecular trees. If there is evidence (even unpublished) for these fossils to be crown group representatives, please explain it in detail in the supplementaries.
And besides, why did you not take the oldest representatives of each group, e.g., H. hennigi from Baltic amber to calibrate Hyptia?
line 245: what is a " 95% highest priority density interval"?

Validity of the findings

I disagree with people that suggest that few-gene analyses are not useful anymore in our time of phylogenomics - the robustness of the molecular data for phylogenetic inference seems to be sufficient here, judging from (most of) the support values. However, the data matrix is far from complete, with some markers only available for about half of the specimens. Please add information on coverage to the main text and not just supplementary material. Also point out clearly which markers were already used in Deans et al. (2006).
Some of these markers were also used in the Hymenoptera tree of life study (Klopfstein et al. 2013) – did you use the same gene portions here, especially for CAD? Also, it is great that you discuss the support in the individual gene trees for certain groupings. Why not add that information graphically to Figure 2 which would increase its value a lot?
Homogeneity tests: It is interesting that you identified significant heterogeneity in some markers, but this apparently did not have an impact on the phylogenetic estimate. Could you discuss that a bit more? Also, in the supplementary, the p values of the X2 test for nucleotide homogeneity are given as "p=0.00". How low were they really? And can a test be "falsified"?
Another point concerns the validity of the choice of calibration densities. This is, however, not only an issue in this paper, but has been discussed for the entire field of node dating for some time (maybe cite Warnock et al. (2011) when discussing sensitivity to node age priors). I like that the authors here perform a sensitivity analysis by using two different choices for the age priors, first a fixed maximum of 25 million years (or rather upper bound of the 95%) and second an approach in which fossils of a more inclusive group are chosen as maximum bounds. The way they then choose the former as the “better” option is however rather ad hoc: “It is likely that the normal analysis uses too broad a range, with the maximum bound being set too far away from the oldest known fossil for the crown lineage” (lines 325 ff.). Please discuss this in more detail. Also, provide a more in-depth justification why the 25-million-year assumption might be a good idea for the Evaniidae and not just for bees. Would a maximum derived from the maximum age of suitable hosts be an alternative for Evaniidae?
You seem to imply that you used a normal instead of a lognormal prior density on the calibrations in your second analysis (lines 318/19 and 321). Did I miss something here? Was it a truncated normal or does it mean you used soft minima also and not just soft maxima? Please make this clear in the MM section.
line 330/31: "consistent with dates for Evanioidea using total-evidence dating (Zhang et al., 2015)." – This reference only included a single (composite) representative of Evanoidea and thus could not possibly be used as a comparison to your estimate of the MRCA of this superfamily. In fact, the analysis that you seem to cite here (diversified sampling results, their Figure 7) was recovering the divergence between a representative of Evanoidea and one of Chalcidoidea as about 80-160 my, which would be in conflict with your (older) age of Evanoidea.
Table S4: why not put the posterior clade probability directly instead of shaded and unshaded "x"es?

·

Basic reporting

The paper is generally very clearly and concisely written, and well organized. I found it very easy to look for answers to questions that popped up, as the use of subtitles/section headings was well designed.

The supporting literature was very well documented - I especially appreciated the section on previous use of each of the genes used in the analysis, and on the characteristics of the new gene.


The manuscript would benefit from decreased use of the word "however" in the results/conclusions, as it tends to disrupt the flow of thought when a simple contrast would be more effective.

Experimental design

The paper used very well-justified and state of the art analytical methods. Some might argue that using only 6 genes for such an analysis is somewhat minimalist in today's context, but the well-chosen genes appear to be very effective in estimating the phylogeny in a convincing manner.

Validity of the findings

I especially like the section where the authors have clearly proposed specific classificatory and nomenclatural changes based on the strong findings of the study - this is often not carried through upon in phylogenetic studies at this level.

I did not find any conclusions that were not supported by the results.

Additional comments

Very nice upgrade to the Deans et al. 2006 study.

Reviewer 3 ·

Basic reporting

The article is clear, professional and self-contained. I had a couple of specific suggestions regarding wording that I will paste below.

Line 16: “Ensign wasps (Hymenoptera: Evaniidae), which develop as predators of cockroach eggs (Blattodea), are
distributed nearly worldwide and exhibit numerous interesting biological phenomena.” - > could you reconfigure this sentence so it wasn’t parenthetic? Something like, ‘Ensign wasps (Hymenoptera: Evaniidae) develop as predators of cockroach eggs (Blattodea) and are distributed nearly worldwide.’
Line 20: “novel marker AM2” -> the marker is not novel, but it’s use in phylogeny estimation is. I would suggest re-phrasing to reflect this. Perhaps something like: “and a marker (gene AM2) used for the first time in phylogenetic reconstruction”.
Line 51: “and biologically compelling “ I would argue for removing this. What taxon isn’t?
Line 59 – “also” is a strange way to start a paragraph. Remove.
Line 254 “according to the authors of the program, they are unlikely to have an effect on the analysis” I’m uncertain if this is the desired form of reference for “pers com” in PeerJ.

Experimental design

The research question is well defined, relevant and meaningful. The data and analyses clearly improve our understanding of Evaniidae systematics. I have three small suggestions below for clarity of reporting.

Line 152 – “has been little explored for phylogenetic studies” – clarify here. Is it little, or none? I didn’t see any pop-sets in NCBI for AM2. On line 336 you say it has never. If never – great, but say it!.
Line 281 “observed notable differences in nucleotide composition across taxa for some genes” – refer here to your Supplemental table. .
Line 344 – “gel cuts” – define or describe here. The specific term doesn’t appear in the Methods.

Validity of the findings

Data is robust, statistically sound and the conclusions are well and concisely stated.

Additional comments

I enjoyed the manuscript and think it will have a good home in PeerJ. I have one general point that I think warrants consideration/alteration before publication.

General Point: COI deletions. The Evania wasp COI here all have in frame 3 or 6 bp deletions. This isn’t abnormal for Hymenoptera (see Hansson et al 2015 and Quicke et al 2012) but it should be noted – particularly in light of your use of molecular motifs such as this in other markers. Also, the COI sequence for Zeuxevania_splendidula_voucher_312 (KY082487.1) has a string of 13 “?” within the sequence near the same region as the Evania deletions. I’m not certain what to make of these “?”. Are these ambiguities? Are they hypothesised deletions? If so – that they are out of frame nature needs to be commented upon. Is this a COI pseudogene?

• Hansson C, Smith MA, Janzen DH, Hallwachs W (2015) Integrative taxonomy of New World Euplectrus Westwood (Hymenoptera, Eulophidae), with focus on 55 new species from Area de Conservación Guanacaste, northwestern Costa Rica. ZooKeys 485: 1-236. https://doi.org/10.3897/zookeys.485.9124
• QUICKE, D. L., ALEX SMITH, M. , JANZEN, D. H., HALLWACHS, W. , FERNANDEZ‐TRIANA, J. , LAURENNE, N. M., ZALDÍVAR‐RIVERÓN, A. , SHAW, M. R., BROAD, G. R., KLOPFSTEIN, S. , SHAW, S. R., HRCEK, J. , HEBERT, P. D., MILLER, S. E., RODRIGUEZ, J. J., WHITFIELD, J. B., SHARKEY, M. J., SHARANOWSKI, B. J., JUSSILA, R. , GAULD[DECEASED], I. D., CHESTERS, D. and VOGLER, A. P. (2012), Utility of the DNA barcoding gene fragment for parasitic wasp phylogeny (Hymenoptera: Ichneumonoidea): data release and new measure of taxonomic congruence. Molecular Ecology Resources, 12: 676-685. doi:10.1111/j.1755-0998.2012.03143.x

---

## Round 0.2 · Major Revisions

Your revised manuscript was reviewed by one of the original reviewers (the other had more minor reservations, and I assessed your revisions myself). As the outstanding reviewer note, nearly all the points have now been satisfactorily addressed.

The only holdout at this point lies with the tree calibration. The reviewer's reservation is justified and she helpfully provides you with a concrete possible ways of resolving this.

Because divergence time is a key aspect of the work, I am tagging this as a "major revision". But I expect that you should be able to address this quite easily, and will strive to handle your revised manuscript as swiftly as possible.

·

Basic reporting

see below

Experimental design

see below

Validity of the findings

see below

Additional comments

The authors have improved the manuscript considerably – I especially like the addition on the biology and the fossil record of the taxon. As for the phylogeny mostly including undescribed species, I still think this is unfortunate, but am of course aware of the scarcity of funding options for alpha taxonomy. I hope the authors will be able to describe these species formally in the future, in order to not further contribute to the gap between accumulating molecular data and proper species names as a reference system.
As to my main point about the setting of calibration points, I am still not entirely convinced that this was done correctly, so I have to get back to my point about crown group versus stem group. For each fossil, you very clearly list the synapomorphies that justify familial or generic placement – that is well done. But as far as I can see (feel free to correct me if I am wrong), this does not exclude the possibility that we are looking at stem group representatives. In some cases, such as the now abandoned Hyptia deansi, there was even indication (presumably plesiomorphic notauli character) that it was indeed a stem group representative. To justify crown group placement, you need to demonstrate that the fossil belongs to a species group WITHIN the genus (same as Archeopterix not being useful for calibrating the MRCA of extant birds, but rather of the ancestor of birds + crocodiles). So, either move (some of) your calibration points one node down or justify crown group placement for each of them.
Minor points:
- Change the title to “divergence TIME estimations”? “Divergence” as such might be misunderstood to mean “sequence divergence”.
- There are still some “to belong for a group” and similar expressions in the supplementary on fossil calibrations, please correct these.

---

## Round 0.3 · accepted · Accept

Thank you for addressing the outstanding points. I indeed agree that the issue of calibration cannot be entirely settled as part of this peer-review process. Meanwhile the issue is appropriately discussed and I am happy to accept your revised manuscript for publication. Since the peer-reviews and your rebuttals contain interesting exchanges on the topic, I encourage you to publish them alongside your manuscript (PeerJ provides this as an option).

#